# The Effects of Oral Semaglutide on Hepatic Fibrosis in Subjects with Type 2 Diabetes in Real-World Clinical Practice: A Post Hoc Analysis of the Sapporo-Oral SEMA Study

**DOI:** 10.3390/ph18010129

**Published:** 2025-01-19

**Authors:** Hiroya Kitsunai, Yuka Shinozaki, Sho Furusawa, Naoyuki Kitao, Miki Ito, Hiroyoshi Kurihara, Chiho Oba-Yamamoto, Jun Takeuchi, Akinobu Nakamura, Yumi Takiyama, Hiroshi Nomoto

**Affiliations:** 1Division of Endocrinology, Metabolism, and Rheumatology, Department of Internal Medicine, Asahikawa Medical University, Asahikawa 078-8510, Hokkaido, Japan; kitsunai@asahikawa-med.ac.jp (H.K.); a210010@ed.asahikawa-med.ac.jp (Y.S.); taka0716@asahikawa-med.ac.jp (Y.T.); 2Department of Rheumatology, Endocrinology and Nephrology, Faculty of Medicine, Graduate School of Medicine, Hokkaido University, Sapporo 060-8638, Hokkaido, Japan; frsw.0008@gmail.com (S.F.); akinbo@huhp.hokudai.ac.jp (A.N.); 3Aoki Clinic, Internal Medicine, Sapporo 003-0023, Hokkaido, Japan; kitao-n@huhp.hokudai.ac.jp; 4Kurihara Clinic, Internal Medicine, Sapporo 004-0053, Hokkaido, Japan; mito@kurihara-naika.net (M.I.); hkurihara@kurihara-naika.net (H.K.); 5Sapporo Thyroid and Diabetes Clinic, Internal Medicine, Sapporo 060-0807, Hokkaido, Japan; chiho@sdtc.jp (C.O.-Y.); jun@sdtc.jp (J.T.)

**Keywords:** GLP-1 receptor agonist, MASLD, type 2 diabetes, semaglutide, observational study

## Abstract

Background/Objectives: Metabolic dysfunction-associated steatotic liver disease (MASLD) is an important common comorbidity in subjects with type 2 diabetes, and liver fibrosis is a factor directly related to its prognosis. Glucagon-like peptide-1 receptor agonists are useful treatment options for MASLD; however, the efficacy of oral semaglutide in treating liver steatosis/fibrosis has not been fully elucidated. Methods: A secondary analysis of a multicenter, retrospective, observational study investigating the efficacy and safety of oral semaglutide in Japanese subjects with type 2 diabetes in a real-world clinical setting (the Sapporo-Oral SEMA study) was conducted. Subjects in the original cohort were divided into groups as follows: subjects with suspected MASLD (alanine aminotransferase > 30 U/L) were placed in an overall group; a subpopulation from an overall group at high risk for hepatic fibrosis (fibrosis-4 (FIB-4) index ≥ 1.3 or platelet count < 200,000/µL) was placed in a high-risk group; and the remaining subjects were placed in a low-risk group. Changes in the hepatic steatosis index and FIB-4 index after oral semaglutide induction were explored using a paired *t*-test or the Wilcoxon signed-rank test. Results: Overall, 169 subjects (including 131 that switched from other medications) were analyzed, and 67 and 102 subjects were selected for the high-risk and low-risk groups, respectively. Oral semaglutide significantly improved the hepatic steatosis index (from 46.1 to 44.6, *p* < 0.001) and FIB-4 index (from 1.04 to 0.96, *p* < 0.001) as well as several metabolic parameters in all cohorts. The efficacy of semaglutide in treating liver fibrosis was confirmed by the addition of, and switching from, existing agent groups. Furthermore, improvement in the FIB-4 index was significantly negatively correlated with the baseline FIB-4 index. Conclusions: The induction of oral semaglutide might be a useful treatment option for subjects with type 2 diabetes at high risk for liver fibrosis, even when switching from conventional medications for diabetes.

## 1. Introduction

Metabolic abnormalities are closely related to impaired liver function, and the deterioration of liver function affects diabetes treatment and prognosis. Non-alcoholic fatty liver disease (NAFLD) is thought to have the potential to progress to liver cirrhosis. Because it has become clear that various metabolic disorders such as obesity, diabetes, and metabolic syndrome are involved in the progression of the disease, a new disease concept termed metabolic dysfunction-associated steatotic liver disease (MASLD) has been proposed [1]. Elastography assessment has shown that approximately 70% of subjects with type 2 diabetes have MASLD, and 21% have liver fibrosis [2]. Non-alcoholic steatohepatitis, an advanced stage of MASLD, termed metabolic dysfunction-associated steatohepatitis (MASH), is the leading cause of progression to cirrhosis and hepatocellular carcinoma [3]. Furthermore, MASLD or MASH, which coexist with diabetes, are significant risk factors for macrovascular diseases such as stroke and coronary heart disease [4], as well as microvascular disorders such as diabetic nephropathy [5], which are related to insulin resistance and low-level inflammation.

Liver biopsy, the gold standard for diagnosing steatotic liver disease or liver fibrosis, is highly invasive [6] and cannot be widely performed. Therefore, several simple and non-invasive scoring systems based on routine clinical test parameters have been proposed and used to screen for fatty liver and liver fibrosis. The serum alanine aminotransferase (ALT) level, fibrosis-4 index (FIB-4), NAFLD fibrosis score (NFS), fatty liver index (FLI), and hepatic steatosis index (HSI) have been used as non-invasive indicators to detect MASLD [7,8]. In 2023, the Japan Society of Hepatology published the “Nara Declaration,” recommending further investigation to evaluate the causes of chronic liver disease in subjects with serum ALT levels above 30 IU/L [9]. In this group of subjects with obesity, diabetes, hypertension, or fatty liver, setting the criteria for the platelet count as <200,000/μL or FIB-4 index ≥ 1.3 may enable the detection of fatty liver with stage 2 or higher fibrosis [10] based on Brunt’s classification [11].

For the treatment of MASLD/MASH, it is necessary to focus on liver dysfunction and to address the symptoms of metabolic syndrome (e.g., hypertension, dyslipidemia, obesity, type 2 diabetes). MASLD increases the risk of cardiovascular disease (CVD) through increased oxidative stress, systemic and hepatic insulin resistance, low-grade inflammation, and metabolic dysfunction, including endothelial dysfunction [12]. Therefore, treatments that improve CVD risk factors (hypertension, dyslipidemia, blood glucose abnormalities, and obesity) might have positive effects on inflammation and fibrosis in MASLD. Recently, several anti-diabetic agents have shown potential to be effective in treating MASLD. Pioglitazone improves insulin resistance by activating peroxisome proliferator-activated receptor gamma; favorably regulates lipid and glucose metabolism; and contributes to reducing portal hypertension, visceral inflammation, angiogenesis, and portosystemic shunting, leading to reduced steatosis and necroinflammation in subjects with MASH [13,14]. Recent guidelines recommend the use of pioglitazone in subjects with diabetes suffering from MASH [15]. Sodium–glucose co-transporter 2 inhibitors (SGLT-2is) have beneficial effects on the cardiovascular system in non-diabetic and diabetic subjects [16]. SGLT2is are expected to contribute to the improvement in MASLD or MASH by improving glycemic control; reducing visceral adiposity, serum uric acid, oxidative stress, and inflammation; and increasing adiponectin [17].

GLP-1 receptor agonists (GLP-1RAs) potently improve glycemic control in a glucose concentration-dependent manner [18]. GLP-1RAs, such as semaglutide, are also recommended for type 2 diabetes subjects at high risk for CVD [19]. Orally administered peptide formulations generally have low bioavailability and are therefore often administered as injections, but oral semaglutide can be absorbed through the gastric mucosa by combining it with the absorption enhancer sodium N-(8-[2-hydroxybenzoyl] amino) caprylate [20]. In the phase 3a PIONEER 4 trial, oral semaglutide was non-inferior to subcutaneous GLP-1RAs at lowering HbA1c and was superior at reducing weight, with safety and tolerability similar to those of subcutaneous GLP-1RAs [21]. However, oral semaglutide must be taken on an empty stomach with up to 120 mL of water, and if not taken properly, administration is complicated and ineffective. Therefore, although oral semaglutide was shown to be effective in clinical trials in which its administration was strictly conducted, its effectiveness in actual clinical practice is unclear. The Sapporo-Oral SEMA study (UMIN000050583) was a multicenter, retrospective, observational study conducted by Furusawa et al. to evaluate the efficacy and safety of oral semaglutide in subjects with type 2 diabetes in a real-world clinical setting [22]. Data from 434 subjects who met the inclusion criteria were analyzed. After a 6-month observation period, subjects were administered oral semaglutide, and glycated hemoglobin and body weight were significantly decreased, with efficacy also confirmed in a subgroup who switched from other antihyperglycemic drugs. Additionally, gastrointestinal symptoms were frequently observed as an adverse event of oral semaglutide, but most subjects could continue semaglutide, and the discontinuation rate was comparable to that in phase III trials. GLP-1RAs improve hyperglycemia as well as the insulin sensitivity of the liver and global/local fat, thereby reducing the amount of lipotoxic metabolites and proinflammatory mediators in the circulation, contributing to hepatoprotection against MASLD and MASH [23]. Although injectable semaglutide has shown efficacy in these conditions [24,25], there is limited knowledge of the different effects of oral semaglutide, which has a different hemodynamic profile compared with the injectable formulation and exerts its effects directly on the liver [20]. The results of the Sapporo-Oral SEMA study showed that oral semaglutide improved glycemic control in real-world clinical settings; however, the relationship with liver disease in real-world clinical settings has not been investigated. Therefore, in the present study, we investigated the effects of oral semaglutide on liver fibrosis in subjects with type 2 diabetes in real-world clinical practice by conducting a post hoc analysis of subject data from the Sapporo-Oral SEMA study.

## 2. Results

### 2.1. Study Population

To clarify the efficacy of the induction of GLP-1RAs on the liver phenotype, 30 of the 434 participants in the original study who had been treated with injectable GLP-1RAs were excluded. Because the purpose of this study was to examine the effects of oral semaglutide on a population with suspected MASLD and with a high risk for liver fibrosis, a subject population was established that was assumed to have steatotic liver disease by excluding cases with ALT < 30 (*n* = 220) based on a recent declaration [9,26]. From this group, 15 subjects for whom the FIB-4 index at baseline and/or 6 months could not be assessed were excluded (a breakdown of the missing data is presented in Appendix A), resulting in an overall cohort of 169 subjects. Consequently, to evaluate whether oral semaglutide was effective against liver damage even in subjects at high risk for liver fibrosis, a high-risk population was extracted using the baseline FIB-4 index and platelet counts based on the same declaration to create a high-risk group (*n* = 67). The remaining 102 subjects were defined as a low-risk group (Figure 1).

As shown in Table 1, the mean age of the overall cohort was 53.0 ± 12.7 years, and the study population consisted of subjects with a relatively high body mass index (BMI) and inadequate HbA1c management under existing treatments for type 2 diabetes in clinical settings. Regarding concomitant medications for diabetes, dipeptidyl peptidase-4 (DPP-4) inhibitors were the most commonly administered, followed by metformin and/or SGLT2is. DPP-4 inhibitors were discontinued in all participants when oral semaglutide was started. Only three subjects were switched from non-DPP-4 inhibitors (one from glinide and two from SGLT2is), whereas the other concomitant medications were continued at the start of oral semaglutide. Thiazolidinedione, an SGLT2i used to treat liver fibrosis, was not used as frequently (4.5% in the high-risk group and 2.9% in the low-risk group) (Table 1). However, most subjects who received thiazolidinedione were also receiving at least three other oral anti-diabetic agents including SGLT2is, suggesting that thiazolidinedione was administered to subjects with worse glycemic control.

### 2.2. Changes in Metabolic Parameters After Oral Semaglutide Induction

Six months after oral semaglutide induction, multiple metabolic parameters showed an overall improvement as expected. The extent of HbA1c reduction was −0.9 (−1.0, −0.7)% in the overall cohort, which was equivalent to that in previously published study data (Table 2). Similar changes in other metabolic indices including BMI, systolic blood pressure, and lipid metabolism were observed. The mean baseline values of hepatobiliary enzymes including aspartate aminotransferase (AST), ALT, and γ-glutamyl transpeptidase (γ-GTP) increased with the progression of hepatic risk from the low-risk group to the high-risk group compared with the values in the original population, and all these values showed a significant improvement in both subpopulations, including the present cohort of subjects selected as at high risk for liver disease (Table 3).

### 2.3. Effects of Oral Semaglutide on Indices for Liver Steatosis and Fibrosis

In this study, the effects of oral semaglutide on 
hepatic steatosis and fibrosis were evaluated using the HSI and FIB-4 index, 
respectively, before and after treatment. At baseline, the HSIs were 46.1 ± 
8.3, 43.7 ± 7.1, and 47.8 ± 8.6 in the overall group, high-risk group, and 
low-risk group, respectively, which were higher than the cutoff values for 
NAFLD (>36) in Japanese subjects [27]. Oral 
semaglutide administration significantly improved HSI scores in all subgroups (*p* 
< 0.001, *p* = 0.004, and *p* < 0.001, respectively), as shown 
in Table 4. The baseline FIB-4 index was 
not as high as 0.86 (0.64–1.01) in the low-risk group but was as high as 1.91 
(1.39–2.69) in the high-risk group, as expected, because subjects at high risk 
for liver fibrosis were selected. After oral semaglutide treatment, the FIB-4 
index in all subgroups was significantly ameliorated (*p* < 0.001 in 
all groups) (Table 4). Because there was 
a difference in the effects of HbA1c reduction between the addition of 
semaglutide and the switch from other oral hypoglycemic agents (OHAs), as shown 
in the original cohort, a secondary analysis of these introduction methods was 
performed. Because subjects switching from existing GLP-1RAs were excluded from 
this study, most subjects were switched from DPP-4 inhibitors in the switching 
group, except for one subject in the high-risk group (switch from glinide) and 
two subjects in the low-risk group (switch from SGLT2is). When oral semaglutide 
was introduced, other medications were continued. Similar changes were observed 
with oral semaglutide in all subgroups, but only the HSI in the switching 
regimen in the high-risk group and the FIB-4 index in the add-on regimen in the 
low-risk group showed no significant changes (Table 4 and Figure 2).

Because the final dose of semaglutide varied from case to case (3 mg/day to 14 mg/day), an analysis of the effects of each dose on the indicators was also performed. As presented in Appendix A, a significant reduction in the HSI was confirmed for all semaglutide doses; however, semaglutide’s effects on the FIB-4 index was not clear in the lowest semaglutide dose (median change from baseline: −0.07, *p* = 0.098).

Finally, we explored changes in the parameters and subject background associated with the improvement in the HSI and FIB-4 index obtained with semaglutide administration using the overall cohort. The improvement in the HSI was significantly correlated with improvements in ALT as expected. In addition, improvements in BMI, HbA1c, and triglyceride correlated with ΔHSI. Regarding the FIB-4 index, only changes in hepatobiliary enzymes correlated with the ΔFIB-4 index (Table 5). Regarding baseline values related to subject background, changes in the HSI were positively correlated with baseline γ-GTP, and changes in the FIB-4 index were negatively correlated with baseline hepatobiliary enzymes and the FIB-4 index (ρ = −0.368, *p* < 0.001), suggesting that the induction of oral semaglutide may be effective for subjects with a high FIB-4 index (Appendix A).

## 3. Discussion

In this post hoc analysis of the Sapporo-Oral SEMA study, we investigated changes in the liver fibrosis index after oral semaglutide induction in type 2 diabetic subjects with suspected MASLD. The results showed that the FIB-4 index was improved in both groups receiving oral semaglutide, either those newly administered oral semaglutide or switched to oral semaglutide from other OHAs (mainly from DPP-4is). This suggests that oral semaglutide may be effective for diabetes treatment and for inhibiting liver fibrosis in subjects with type 2 diabetes with suspected MASLD. Oral semaglutide also reduced the FIB-4 index in subjects with type 2 diabetes at high risk for liver fibrosis. Moreover, an interesting result of this analysis was that the change in the FIB-4 index was negatively correlated with the baseline FIB-4 index. This suggests that oral semaglutide may have a stronger effect on subjects with more severe liver fibrosis.

Our correlation analysis showed no association between changes in the FIB-4 index and baseline HbA1c or changes in HbA1c, suggesting that factors other than glycemic control may be involved in the inhibition of liver fibrosis via oral semaglutide. Chronic liver diseases such as MASLD are characterized by persistent inflammation and subsequent liver fibrosis [28]. GLP-1RAs suppress various inflammatory markers. Studies in rat and mouse models reported that exogenous GLP-1RA administration reduced the blood concentrations of proinflammatory cytokines, including interleukin (IL)-1α, IL-1β, IL-6, tumor necrosis factor-alpha (TNFα), interferon-gamma (IFNγ), and transforming growth factor-beta (TGFβ) [29,30]. Several meta-analyses reported that GLP-1RAs reduced inflammatory biomarkers, such as C-reactive protein (CRP) and TNFα, and oxidative stress biomarkers, such as malondialdehyde, in subjects with type 2 diabetes [31,32]. Another mechanism is the progression of liver fibrosis due to the overproduction and accumulation of the extracellular matrix produced by activated hepatic stellate cells differentiating into myofibroblasts [33]. GLP-1RAs were reported to inhibit hepatic stellate cell activation by inhibiting the p38 MAPK signaling pathway [34]. However, oral semaglutide also reduced the HSI in subjects with type 2 diabetes with suspected MASLD. Interestingly, unlike the FIB-4 index, the change in the HSI was positively correlated with the change in HbA1c. In diabetic subjects, the HSI and FLI were reported to be associated with HbA1c levels independently of body weight and diabetes treatment [35,36], and HbA1c was also associated with the severity of fatty liver assessed by liver biopsy [37]. The presence of hyperglycemia and insulin resistance is known to promote fibrogenesis by inducing the expression of insulin and insulin-like growth factor 1 receptors and advanced glycation end products in hepatic stellate cells and upregulating TGFβ and connective tissue growth factor (CTGF) [38]. Another finding from this study is that switching from existing OHAs, primarily DPP-4 inhibitors, which are also incretin-related drugs, was also effective at improving liver fibrosis. Considering that previous studies failed to demonstrate the presence of GLP-1 receptors in hepatocytes [39,40], the potent hepatic lipogenesis/fibrosis ameliorating effects of oral semaglutide may be associated with improved systemic metabolism, including glycemic management and weight loss [14]. Thus, oral semaglutide is expected to prevent hepatic steatosis by improving glycemic control and suppress liver fibrosis by inhibiting the activation of hepatic stellate cells.

This study had several limitations as previously described [22], including being a single-arm, retrospective, open-label study, with a lack of assessment of adherence to oral semaglutide, no restrictions on medication changes for comorbid conditions, and variations in baseline treatment regimens making it difficult to conclusively compare the efficacy of oral semaglutide with specific pretreatment medications. In addition, because this study was retrospective, it was not sufficiently adjusted for various confounding factors. During the evaluation of liver fibrosis using the FIB-4 index, the age of the subject can be a confounding factor [41], and a cutoff value of the FIB-4 index ≥2.0 has also been proposed for subjects aged ≥65 years old. We investigated the changes in the hepatic steatosis index (HSI) and FIB-4 index with oral semaglutide administration in 15 subjects aged ≥65 years old with a FIB-4 index ≥2.0. As shown in Appendix A, there was no change in the HSI with oral semaglutide, but the FIB-4 index was significantly decreased. In addition, no investigation was conducted on the drinking history of the target subjects; therefore, its influence on the results was not examined. Furthermore, this study did not perform a histological evaluation using liver biopsy or a numerical liver stiffness evaluation using magnetic resonance elastography [42] or transient elastography [43], making it difficult to confirm the correlation between the actual degree of liver fibrosis and oral semaglutide, which remains to be shown in future studies. However, the strengths of this study include having a sample size sufficiently large to allow for a highly reliable statistical analysis and the analysis of real-world clinical practice. This study highlighted the effects of oral semaglutide on subjects with liver fibrosis under routine medical care.

## 4. Materials and Methods

### 4.1. Study Design and Participants

This was a secondary analysis of our previous multicenter, retrospective, observational study investigating the efficacy and safety of oral semaglutide in Japanese subjects with type 2 diabetes in a real-world clinical setting (the Sapporo-Oral SEMA study) [22]. Briefly, data on all subjects with type 2 diabetes aged 20 years or older attending the collaborating institutions who were started on oral semaglutide between February 2021 and December 2022 were extracted from their medical records, and changes in metabolic indices, including changes in HbA1c from 6 months before to 6 months after semaglutide initiation, and the safety of oral semaglutide were evaluated. No drugs were restricted for use at the time of inclusion, including the administration of injectable GLP-1RAs. Exclusion criteria were subjects (1) who had an allergy to semaglutide, (2) experienced diabetic ketoacidosis and/or severe infections, or (3) were incompatible with the trial for other reasons (as determined by physicians). At this stage, subjects with acute liver injury, active viral chronic hepatitis, or liver cancer were excluded.

In this secondary analysis, we focused on the efficacy of oral semaglutide in treating liver steatosis and/or fibrosis. Therefore, we referred to the recently proposed recommendation of the Japan Society of Hepatology (Nara Declaration) [9,26] and extracted high-risk populations from our original data. As the first step, this analysis excluded switching from existing injectable GLP-1RAs other than oral semaglutide to clearly assess the effects of the introduction of oral semaglutide on the liver. Subsequently, the remaining cases with ALT >30 U/L were selected from the cohort based on the Nara Declaration [26], and a high-risk population was established for steatotic liver disease. From these, a group of cases for which pre- and post-FIB-4 index comparisons were not possible was excluded and designated as the overall cohort in this study (Figure 1). To extract a steatotic liver population at higher risk for liver fibrosis, we defined a high-risk group by selecting cases from the overall cohort with a platelet count of 200,000/µL and an FIB-4 index of 1.3 as cutoff values, and the remaining subjects were defined as a low-risk group (Figure 1).

As shown in the original study, oral semaglutide was initiated at 3 mg once a day for at least 4 weeks, followed by escalation to 7 mg after at least 4 weeks, and then scaled up to 14 mg based on the physicians’ judgment, if necessary. The subgroups were classified according to the method of the introduction of oral semaglutide: “Add-on group” with a new additional dose and “Switch from other OHAs” group with a switch from one existing OHA. Thus, subjects taking DPP-4 inhibitors were in the switch group because oral semaglutide was started instead of DPP-4 inhibitors in all cases. The changes in medications for diabetes and concomitant comorbidities were not restricted because of the retrospective study design.

Opt-out consent provision was adopted in this retrospective study based on the Ethics Committee’s formulation, and the study protocol was approved by the Institutional Review Board of the Japan Clinicians Diabetes Association and Hokkaido University (approval number 022-0200). This study was conducted in accordance with the principles of the Declaration of Helsinki and its amendments.

### 4.2. Evaluation Method of Liver Steatosis and Fibrosis Indices Using Surrogate Markers

In this study, blood tests performed before and 6 months after oral semaglutide administration were used to assess the extent of liver steatosis and fibrosis in the participating subjects, using the surrogate markers the HSI and FIB-4 index. These variables were calculated using the following formulas: HSI = 8 × [ALT/AST ratio] + BMI (+2, if female; +2, if diabetes mellitus) [44], and FIB-4 index = age × [AST/(platelet count × ALT)^1/2^] [45]. The primary endpoint of this subanalysis was changes in these indices from the baseline. As secondary endpoints, changes in physical assessment including BMI and metabolic parameters after oral semaglutide induction were analyzed. In addition, clinical parameters that correlated with an improvement in the HSI and FIB-4 index were explored.

### 4.3. Statistical Analysis

Data were presented as the mean ± SD, median (25% percentile, 75% percentile) for continuous variables, or *n* (%) for categorical variables. The mean changes from the baseline to the end of this study were expressed as the mean or median (95% confidence interval). Data within the groups were compared using a paired *t*-test or the Wilcoxon signed-rank test. Comparisons between groups were assessed using an unpaired *t*-test or Mann–Whitney *U*-test for continuous variables. Correlations between two variables were evaluated by Spearman’s rank correlation analysis. Data were analyzed using JMP Pro v17.0.0 (SAS Institute, Cary, NC, USA) and GraphPad Prism 8 v8.4.2 (GraphPad Software, Inc. San Diego, CA, USA). *p*-values < 0.05 were considered statistically significant.

## 5. Conclusions

Oral semaglutide significantly improved indices reflecting liver steatosis/fibrosis in subjects with type 2 diabetes suspicious for MASLD in real-world clinical settings. The FIB-4 index improved significantly in both add-on and switching treatment regimens, especially in subjects with higher baseline values. Although further studies including histological and imaging assessments are needed, the induction of oral semaglutide might be a useful treatment option for subjects with type 2 diabetes at high risk for liver fibrosis, even when switching from conventional medications for diabetes.

## Figures and Tables

**Figure 1 pharmaceuticals-18-00129-f001:**
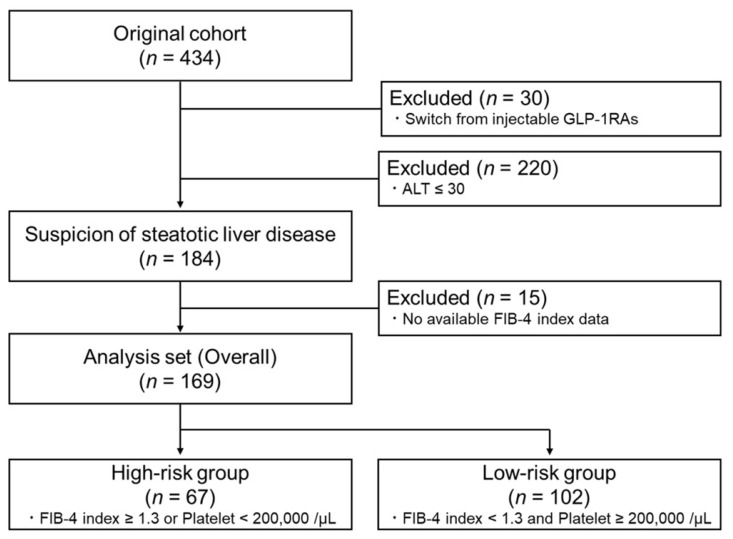
A flow diagram of the subanalysis. From the original cohort, an overall cohort was created by excluding cases treated with injectable GLP-1RAs, with low ALT, or missing FIB-4 index measurements. A high-risk group was selected from this population with a higher risk for liver fibrosis progression, and the remaining subjects were defined as a low-risk group. GLP-1RA, glucagon-like peptide-1 receptor agonist; ALT, alanine aminotransferase; FIB-4, fibrosis-4.

**Figure 2 pharmaceuticals-18-00129-f002:**
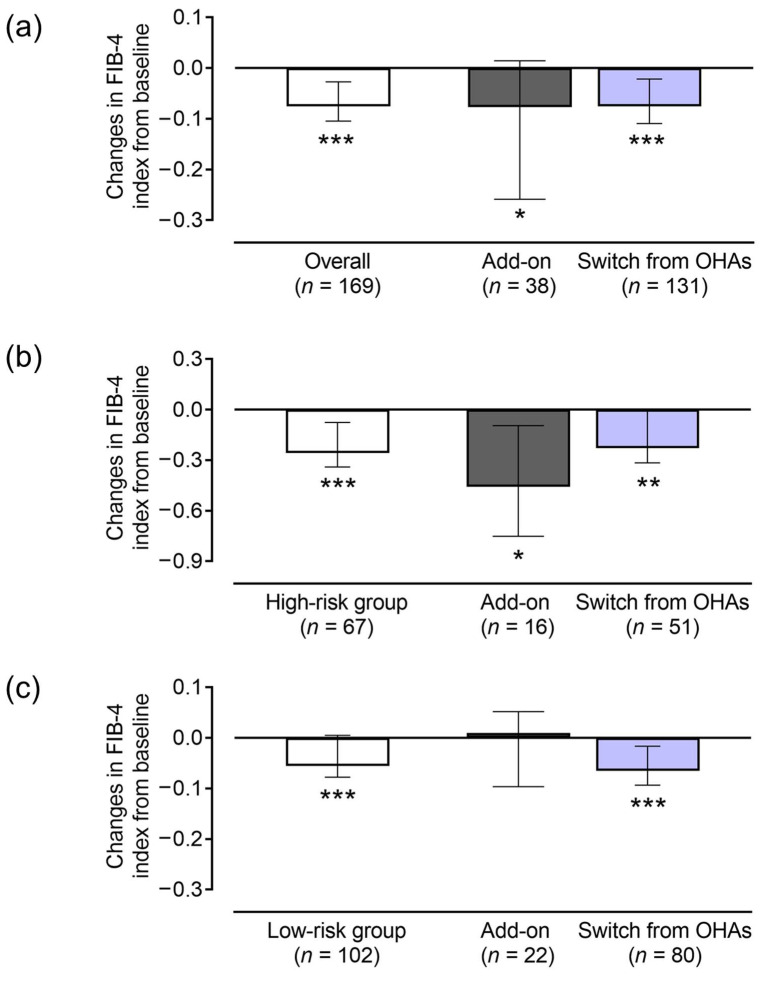
Changes in FIB-4 index after oral semaglutide treatment using treatment regimen. (**a**) Analyses of overall cohort, (**b**) high-risk group, and (**c**) low-risk group. Bars represent median changes from each baseline (95% confidence interval). * *p* < 0.05, ** *p* < 0.01, and *** *p* < 0.001 vs. baseline. FIB-4, fibrosis-4; OHA, oral hypoglycemic agent.

**Table 1 pharmaceuticals-18-00129-t001:** Demographic and clinical characteristics of study participants at baseline.

Variables	Overall (*n* = 169)	High-Risk Group (*n* = 67)	Low-Risk Group (*n* = 102)	*p*-Values (High Risk vs. Low Risk)
Age (years)	53.0 ± 12.7	59.3 ± 10.9	48.8 ± 12.1	<0.001
Female sex (*n*, %)	69 (40.8)	35 (52.2)	34 (33.3)	0.017
Duration of diabetes, *n* (%)				0.555
<5 years	55 (32.5)	18 (26.9)	37 (36.3)	
5 to 10 years	34 (20.1)	15 (22.4)	19 (18.6)	
10 to 15 years	32 (18.9)	15 (22.4)	17 (16.7)	
15≤ years	48 (28.4)	19 (28.4)	29 (28.4)	
HbA1c (%)	7.8 ± 1.1	7.7 ± 1.0	7.9 ± 1.2	0.313
Body weight (kg)	86.2 ± 21.6	82.8 ± 21.0	88.5 ± 21.8	0.093
Body mass index (kg/m^2^)	31.4 ± 7.0	30.9 ± 6.5	31.7 ± 7.3	0.477
Number of antihyperglycemic agents	2.6 ± 1.2	2.5 ± 1.3	2.6 ± 1.1	0.802
Types of antihyperglycemic agents, *n* (%)				
DPP-4 inhibitors	128 (75.7)	50 (74.6)	78 (76.5)	0.855
Metformin	122 (72.2)	47 (70.1)	75 (73.5)	0.726
SGLT2 inhibitors	118 (69.8)	47 (70.1)	71 (69.6)	1.000
Sulfonylureas	29 (17.2)	12 (17.9)	17 (16.7)	0.838
Glinides	16 (9.5)	7 (10.4)	9 (8.8)	0.791
Alfa-glycosidase inhibitors	12 (7.1)	2 (3.0)	10 (9.8)	0.128
Thiazolidinediones	6 (3.6)	3 (4.5)	3 (2.9)	0.682
Insulin injection	5 (3.0)	3 (4.5)	2 (2.0)	0.386

Data are shown as the mean ± SD or number (%). *p*-values between the high-risk group and low-risk group were obtained using an unpaired *t*-test or Fisher’s exact test. HbA1c, glycated hemoglobin; DPP-4, dipeptidyl peptidase-4; SGLT2, sodium–glucose co-transporter 2.

**Table 2 pharmaceuticals-18-00129-t002:** Changes in clinical parameters during the study period.

Variables	Overall (*n* = 169)		
	Baseline	Changes	*p*-Values
Body weight (kg) ^†^	86.2 ± 21.6	−3.0 (−3.5, −2.4)	<0.001
BMI (kg/m^2^) ^†^	31.4 ± 7.0	−1.1 (−1.3, −0.9)	<0.001
SBP (mmHg) ^††^	133.5 ± 16.9	−6.2 (−8.8, −3.6)	<0.001
DBP (mmHg) ^††^	81.2 ± 11.9	−1.9 (−3.4, −0.4)	0.015
HbA1c (%) ^†^	7.8 ± 1.1	−0.9 (−1.0, −0.7)	<0.001
AST (U/L)	35.0 (27.0–50.5)	−4.0 (−7.0, −3.0)	<0.001
ALT (U/L)	49.0 (37.0–71.5)	−8.0 (−11.0, 5.0)	<0.001
γ-GTP (U/L)	54.0 (34.0–83.5)	−8.0 (−10.0, −6.0)	<0.001
eGFR (mL/min/1.73 m^2^)	83.6 ± 21.8	−3.0 (−4.4, −1.6)	<0.001
UACR (mg/g·Cre) ^†††^	22.1 (8.4–55.1)	−3.7 (−5.7, −1.0)	<0.001
HDL-C (mg/dL) ^†^	54.1 ± 13.5	0.3 (−1.0, 1.5)	0.662
LDL-C (mg/dL) ^††^	97.0 ± 25.3	−5.9 (−9.4, −2.3)	0.001
Triglyceride (mg/dL)	145 (106–223)	−8.0 (−16.0, −2.0)	0.010

Data are shown as the mean ± SD and median (25–75%) for the baseline and mean or median (95% confidence interval) for changes from the baseline. *p*-values were obtained using a paired *t*-test or Wilcoxon test. ^†,††,†††^ Data were obtained from 168, 165, and 151 participants in the overall group, respectively. BMI, body mass index; SBP, systolic blood pressure; DBP, diastolic blood pressure; HbA1c, glycated hemoglobin; AST, aspartate aminotransferase; ALT, alanine aminotransferase; γ-GTP, γ-glutamyl transpeptidase; eGFR, estimated glomerular filtration rate; UACR, urinary albumin–creatinine ratio; HDL-C, high-density lipoprotein cholesterol; LDL, low-density lipoprotein cholesterol.

**Table 3 pharmaceuticals-18-00129-t003:** Changes in clinical parameters during study period stratified by risk of liver fibrosis at baseline.

	High-Risk Group (*n* = 67)		Low-Risk Group (*n* = 102)		*p*-Values Between Groups
	Baseline	Changes	Baseline	Changes	
Body weight (kg) ^†,‡‡^	82.8 ± 21.0	−3.2 (−4.2, −2.2) ***	88.5 ± 21.8	−2.8 (−3.5, −2.1) ***	0.537
BMI (kg/m^2^) ^†,‡‡^	30.9 ± 6.5	−1.2 (−1.6, −0.8) ***	31.7 ± 7.3	−1.0 (−1.3, −0.8) ***	0.396
SBP (mmHg) ^†††,‡‡^	133.3 ± 18.2	−6.8 (−11.2, −2.4) **	133.6 ± 16.2	−5.8 (−9.1, −2.5) ***	0.700
DBP (mmHg) ^†††,‡‡^	78.7 ± 13.1	−0.4 (−2.7, 1.9)	82.9 ± 10.8	−2.8 (−4.8, −0.8) **	0.125
HbA1c (%) ^‡^	7.7 ± 1.0	−0.9 (−1.1, −0.7) ***	7.9 ± 1.2	−0.9 (−1.1, −0.7) ***	0.820
AST (U/L)	45.0 (35.0–61.0)	−9.0 (−13.0, −5.0) ***	30.0 (25.0–38.5)	−3.0 (−4.0, −2.0) ***	0.026
ALT (U/L)	49.0 (39.0–73.0)	−9.0 (−16.0, −5.0) ***	47.5 (36.0–66.3)	−6.0 (−11.0, −4.0) ***	0.191
γ-GTP (U/L)	66.0 (39.0–113.0)	−10.0 (−15.0, −7.0) ***	48.0 (30.8–73.5)	−6.0 (−10.0, −4.0) ***	0.250
eGFR (mL/min/1.73 m^2^) ^‡^	78.3 ± 20.9	−3.2 (−5.9, −0.4) *	86.7 ± 21.7	−3.2 (−4.5, −1.4) ***	0.880
UACR (mg/g·Cre) ^††††,‡‡‡^	27.7 (9.4–79.4)	−4.4 (−6.1, −1.0) *	18.9 (8.3–54.5)	−3.2 (−6.7, −0.5) **	0.982
HDL-C (mg/dL) ^‡^	56.3 ± 14.4	1.4 (−0.9, 3.6)	52.6 ± 12.8	−0.4 (−1.9, 1.1)	0.170
LDL-C (mg/dL) ^††,‡‡^	89.7 ± 26.9	−5.7 (−10.3, −1.0) *	101.7 ± 23.0	−6.0 (−11.1, −0.9) *	0.925
Triglyceride (mg/dL)	138 (96–205)	−5.0 (−14.0, 7.0)	150 (111–229)	−12.0 (−21.0, −2.0) *	0.428

Data are shown as the mean ± SD and median (25–75%) for the baseline and mean or median (95% confidence interval) for changes from the baseline. * *p* < 0.05, ** *p* < 0.01, and *** *p* < 0.001 vs. each baseline (paired *t*-test or Wilcoxon test). *p*-values between the groups were obtained using an unpaired *t*-test or Mann–Whitney U-test. ^†,††,†††,††††^ Data were obtained from 66, 65, 62, and 52 participants in the high-risk group, respectively. ^‡,‡‡,‡‡‡^ Data were obtained from 101, 100, and 86 participants in the low-risk group, respectively. BMI, body mass index; SBP, systolic blood pressure; DBP, diastolic blood pressure; HbA1c, glycated hemoglobin; AST, aspartate aminotransferase; ALT, alanine aminotransferase; γ-GTP, γ-glutamyl transpeptidase; eGFR, estimated glomerular filtration rate; UACR, urinary albumin–creatinine ratio; HDL-C, high-density lipoprotein cholesterol; LDL, low-density lipoprotein cholesterol.

**Table 4 pharmaceuticals-18-00129-t004:** Changes in indices for liver steatosis and liver fibrosis.

Variables	Hepatic Steatosis Index			FIB-4 Index		
	Baseline	6 Months	*p*-Values	Baseline	6 Months	*p*-Values
Overall						
Total (*n* = 169)	46.1 ± 8.3	44.6 ± 8.8	<0.001	1.04 (0.78–1.55)	0.96 (0.69–0.96)	<0.001
Add-on (*n* = 38)	47.7 ± 7.7	45.7 ± 8.3	0.002	1.03 (0.73–1.48)	1.01 (0.70–1.30)	0.023
Switch from other OHAs (*n* = 131)	45.7 ± 8.4	44.3 ± 9.0	<0.001	1.05 (0.80–1.58)	0.96 (0.67–1.47)	<0.001
High-risk group						
Total (*n* = 67)	43.7 ± 7.1	42.4 ± 7.1	0.004	1.91 (1.39–2.69)	1.56 (1.21–2.24)	<0.001
Add-on (*n* = 16)	48.4 ± 8.3	45.7 ± 8.1	0.029	1.67 (1.38–2.05)	1.25 (1.08–1.64)	0.013
Switch from other OHAs (*n* = 51)	42.1 ± 6.0	41.3 ± 6.4	0.063	1.94 (1.42–2.83)	1.82 (1.30–2.52)	0.008
Low-risk group						
Total (*n* = 102)	47.8 ± 8.6	46.1 ± 9.6	<0.001	0.86 (0.64–1.01)	0.76 (0.57–0.94)	<0.001
Add-on (*n* = 22)	47.2 ± 7.4	45.7 ± 8.7	0.032	0.86 (0.68–1.01)	0.81 (0.66–1.02)	0899
Switch from other OHAs (*n* = 80)	47.9 ± 8.9	46.2 ± 9.8	<0.001	0.86 (0.61–1.00)	0.74 (0.56–0.92)	<0.001

Data are shown as the mean ± SD or median (25–75%). *p*-values were obtained using a paired *t*-test or Wilcoxon test. OHAs, oral hypoglycemic agents.

**Table 5 pharmaceuticals-18-00129-t005:** The relationship between changes in indices for liver steatosis/fibrosis and other metabolic parameters pre- and post-treatment with oral semaglutide in the overall cohort.

Variables	Changes in HSI		Changes in FIB-4 Index	
	ρ	*p*-Values	ρ	*p*-Values
ΔBody mass index	0.443	<0.001	0.122	0.119
ΔHbA1c	0.274	<0.001	0.089	0.250
ΔAST	0.032	0.680	<0.001	0.734
ΔALT	0.523	<0.001	0.304	<0.001
Δγ-GTP	0.117	0.134	0.314	<0.001
ΔTriglyceride	0.325	<0.001	−0.032	0.684
ΔHDL cholesterol	−0.023	0.769	−0.149	0.054
ΔLDL cholesterol	0.102	0.198	0.054	0.489

Data were analyzed using Spearman’s rank correlation. HbA1c, glycated hemoglobin; AST, aspartate aminotransferase; ALT, alanine aminotransferase; γ-GTP, γ-glutamyl transpeptidase; HDL, high-density lipoprotein; LDL, low-density lipoprotein.

## Data Availability

The data that support the findings of this study are available from the corresponding author upon reasonable request.

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
