# Peer review of "The Effects of Oral Semaglutide on Hepatic Fibrosis in Subjects with Type 2 Diabetes in Real-World Clinical Practice: A Post Hoc Analysis of the Sapporo-Oral SEMA Study"

_pharmaceuticals, 2025, doi:10.3390/ph18010129_

Round 1

Reviewer 1 Report

Comments and Suggestions for Authors

The study addresses an important co-morbidity of diabetes, metabolic dysfunction–associated steatotic liver disease (MASLD), and the ameliorative effect of oral semaglutide.  The oral formulation of semaglutide ensures that it reaches the liver before circulation to the rest of the body, and the suggestion is that oral administration may make semaglutide more effective in improving liver condition, compared with administration of semaglutide by injection.  The study involves patients in a real-world clinical setting, and consequently there is no control group.  End-point analysis focusses on monitoring changes of biomarkers and clinical parameters within patients longitudinally. Patients with potential for MASLD are selected from a larger cohort from a study on the effectiveness or oral semaglutide in the treatment of type 2 diabetes (T2D), which had been published previously [ref 22].

The study data are abundant, and include many important biomarkers for T2D and liver function, presented clearly.  The findings indicate that changing previous therapies for T2D to oral semaglutide can improve (or sustain an ongoing improvement?) in liver condition over a period of 6 months.  This is a useful finding and suggests that oral forms of semaglutide have clinical potential.

One aspect of the study design that is puzzling is the selection of cohorts.  Cohort 2 is effectively a subset of cohort 1, but is analysed as though it is an independent group.  Cohort 1 represents all patients with a propensity for liver disease (MASLD) and cohort 2 represents those within cohort 1 with a high risk for liver fibrosis.  It is useful to identify cohort 2 patients, but why not compare them with the remainder of cohort 1, representing patients with MASLD but are low risk for fibrosis?

Some points to address:

Introduction

Line 85: "GLP-1RAs, such as SGLT2is, are also recommended for type 2 diabetes..." SGLT2is should be replaced by the name of a GLP-1RA e.g. semaglutide?

Methods

The description of allocation to cohorts is clear, but see comment above. Comparing these cohorts e.g lines 170-186 does not make sense because they are not independent sets of data.  It makes more sense to have Cohort 1 divided into Low-risk and High-risk cohorts for fibrosis. Similarly line 157 refers to "the progression of hepatic risk from Cohort 1 to Cohort 2" - the progression would be more distinct if the cohorts were separate.

The issue of concomitant medications is not described clearly.  Line 300: is the "Add-on" group already receiving i.m. semaglutide or insulin?  How many in the group - they are not listed in Table 1? Was it 38, as suggested in Table 3?  Similarly with the switch group - according to Table 1, a total of 431 oral antihyperglycemic agents were given amongst 169 patients of cohort 1, 168 OHAs for 69 patients of cohort 2.  Were all these OHAs stopped and replaced by oral semaglutide?  Given that this article is aimed at a pharmaceutical audience, this point should be clarified.

Results

Table 4.  Should state which cohort of patients are analysed. Described as "changes in the parameters and patient background associated with the improvement in HSI and FIB-4 index obtained with semaglutide administration" is this cohort 1 or cohort 2?

Reference is made to Figure S1 (line 197) and Table S1 (line 209), but these were not provided.

Overall

Two issues need addressing before publication can be recommended.

1. Justification of cohort design, or preferably comparison of Low and High risk fibrosis cohorts.  

2. Much more description of the changes in therapy, and more emphasis of these in the discussion, conclusion and abstract.  It's not clear how well the oral semaglutide performs in comparison with previous treatments.  Is this what is meant by "the original cohort"  in line 153?  This whole issue needs clarification.

Reviewer 2 Report

Comments and Suggestions for Authors

The manuscript addresses the use of semaglutide in an interventionary study.

I think the work is novel, well presented and justified. I think it can be accepted for publication provided some clarifications and performance s are previously carried out.

I would mention the following:

Abstract

Line 24: This ? Please, perform.

Indicate the statistical significance. Also, the statistical design carried out.

Keywords

Include semaglutide and interventionary study.

Introduction

Line 114: Include: “In the present study, …”.

Results

Table 2: Indicate the number of replicates for obtaining the mentioned SD values. Also in Table 3 and Figure 2.

Reviewer 3 Report

Comments and Suggestions for Authors

I have carefully studied the manuscript entitled "The Effect of Oral Semaglutide on Hepatic Fibrosis in Subjects with Type 2 Diabetes in Real-World Clinical Practice: Post-Hoc Analysis of the Sapporo-Oral SEMA Study" by KitsunaiH. et al.

The core idea of the manuscript is intriguing, focusing on wether semaglutide, in the form of per os treatment, could potentially offer some advantage to patients with type-2 diabetes and increased markers for liver fibrosis.

The text is easy to understand; the language used is devoid of major typos and syntax errors. However, before considering publication, the authors are kindly invited to assess / discuss the following issues: 

Major issues

1) Line 125: The authors report that "From this group, 15 patients for whom the FIB-4 index could not be assessed were excluded". This immediatly poses the question regarding the underlying reasons which lead to this exclusion. In other words, which of the four components of FIB-4 was lacking, thus leading to the inability of FIB-4 assessment?

2) Line 133 (Figure 1): The authors report that 102 patients were excluded due to FIB-4 index <1.3. However, it has been demonstrated that age is a confounding factor for the accurate non-invasive diagnosis of fibrosis (see: McPherson S, Hardy T, Dufour JF, Petta S, Romero-Gomez M, Allison M, Oliveira CP, Francque S, Van Gaal L, Schattenberg JM, Tiniakos D, Burt A, Bugianesi E, Ratziu V, Day CP, Anstee QM. Age as a Confounding Factor for the Accurate Non-Invasive Diagnosis of Advanced NAFLD Fibrosis. Am J Gastroenterol. 2017 May;112(5):740-751. doi: 10.1038/ajg.2016.453. Epub 2016 Oct 11. PMID: 27725647; PMCID: PMC5418560). As a consequence, a FIB-4 index cutoff of 2.0 has been proposed for patients aged >65 years. The authors are kindly suggested to i) clarify and discuss this issue, ii) apply the age-specific use of FIB-4 in their data.

3) Line 144: The authors report that "Thiazolidinedione, which has as strong evidence as SGLT2is for liver fibrosis, was not used as frequently (3.6% in Cohort 1 and 4.5% 145 in Cohort 2)". The authors are kindly indited to further discuss the scarcity of  the use of pioglitazone among patients prone to liver fibrosis, focusing on the representativeness of the sample.

4) Line 147 (Table 1): The authors are kindly suggested to provide p-values, thus facilitating the detection of potential confounding factors.

5) Line 161 (Table 2): The authors are kindly suggested to dissect the between-samples variability from the within-samples variability by implementing an appropriate statistical test (e.g. Repeated Measures GLM). 

6) Line 187 (Table 3): In keeping with the previous comment, the authors are kindly suggested to dissect the between-samples variability from the within-samples variability by implementing an appropriate statistical test (e.g. Repeated Measures GLM). 

Round 2

Reviewer 1 Report

Comments and Suggestions for Authors

Thank you for addressing the concerns raised and for recalculating the data as suggested.  The clarity of the script was also improved by providing more details about the OHAs and changes in treatment.

Reviewer 2 Report

Comments and Suggestions for Authors

The manuscript has been performed according to previous comments. I think it can be accepted for publication.

Reviewer 3 Report

Comments and Suggestions for Authors

I have carefully studied the revised version of manuscript entitled "The Effect of Oral Semaglutide on Hepatic Fibrosis in Subjects with Type 2 Diabetes in Real-World Clinical Practice: Post-Hoc Analysis of the Sapporo-Oral SEMA Study" by KitsunaiH. et al.

The authors have successfully responded to every query / issue raised during the review process. As a consequence, the quality of the manuscript has been substantially ameliorated. There are no additional comments, therefore the revised manuscript could merit publication as is.